# Bioinformatic miRNA-mRNAs Analysis Revels to miR-934 as a Potential Regulator of the Epithelial–Mesenchymal Transition in Triple-Negative Breast Cancer

**DOI:** 10.3390/cells12060834

**Published:** 2023-03-08

**Authors:** Jorge Alberto Contreras-Rodríguez, Jonathan Puente-Rivera, Diana Margarita Córdova-Esparza, Stephanie I. Nuñez-Olvera, Macrina Beatriz Silva-Cázares

**Affiliations:** 1Facultad de Informática, Universidad Autónoma de Querétaro, Querétaro 76010, Mexico; 2División de Investigación, Hospital Juárez de México, Ciudad de México 07760, Mexico; 3Departamento de Biología Celular y Fisiología, Instituto de Investigaciones Biomédicas, Universidad Nacional Autónoma de México, Ciudad de México 04510, Mexico; 4Coordinación Académica Región Altiplano, Universidad Autónoma de San Luis Potosí, San Luis Potosí 78300, Mexico

**Keywords:** miR-934–PTEN–EGR2 axis, mesenchymal–epithelium transition, triple-negative breast cancer, miRNAs

## Abstract

Triple-negative breast cancer (TNBC) is one of the most aggressive subtypes of breast cancer and has the worst prognosis. In patients with TNBC tumors, the tumor cells have been reported to have mesenchymal features, which help them migrate and invade. Various studies on cancer have revealed the importance of microRNAs (miRNAs) in different biological processes of the cell in that aberrations, in their expression, lead to alterations and deregulations in said processes, giving rise to tumor progression and aggression. In the present work, we determined the miRNAs that are deregulated in the epithelial–mesenchymal transition process in breast cancer. We discovered that 25 miRNAs that regulate mesenchymal genes are overexpressed in patients with TNBC. We found that miRNA targets modulate different processes and pathways, such as apoptosis, FoxO signaling pathways, and Hippo. We also found that the expression level of miR-934 is specific to the molecular subtype of the triple-negative breast cancer and modulates a set of related epithelial–mesenchymal genes. We determined that miR-934 inhibition in TNBC cell lines inhibits the migratory abilities of tumor cells.

## 1. Introduction

Triple-negative breast cancer (TNBC) is one of the most aggressive subtypes of breast cancer (BC) and is characterized by the lack of expression of the estrogen receptor (ER), the progesterone receptor (PR), and receptor 2 of the human epidermal growth factor (HER2). This subtype represents 15% of BCs and is defined as a heterogeneous group of breast tumors due to the tumors’ diverse histological, genomic, and clinical characteristics and different responses to therapy [1,2]. TNBC tumors present mesenchymal and metastatic characteristics and are correlated with high mortality, poor prognosis, and resistance to therapies [3,4,5].

Epithelial–mesenchymal transition (EMT) is a physiological process involved in embryogenesis and wound healing. However, in pathological conditions such as cancer, this process contributes to the initiation, progression, invasion, and metastasis of tumor cells [4,6,7]. EMT is described as a dynamic and reversible process in which immobile epithelial cells gain mesenchymal characteristics that bestow on them mobile and invasive capabilities due to poor cell adhesion; loss of apical–basal polarity; the degradation of the basal extracellular matrix promoted by the increased expression of proteolytic enzymes, such as matrix metalloproteinases (MMPs), serine proteases, cysteine proteases, disintegrin, and ADAM metalloproteinase [8]. In addition, activation of transcriptional factors, such as omeobox 1 (ZEB1) and ZEB2, snail (SNAI1), slug (SNAI2), and twist-related protein 1 (TWIST1), promotes the expression of mesenchymal markers such as vimentins, N-cadherin, α-actin, and fibronectin, which negatively regulates the expression of epithelial markers such as E-cadherin, tight junction proteins, and cytokeratin [9].

The homologous tumor suppressor of phosphatase and tensin (PTEN) downregulation is common in TNBC. Accordingly, previous investigations have reported a relationship between PTEN expression and the EMT process [10,11,12,13]. In contrast, some studies have suggested that, in different cancer types, early growth response 2 (EGR2) expression is correlated with apoptosis promotion, with the ETM process inhibitor playing a role in this correlation. However, the role of PTEN in TNBC is currently unclear [14].

Since their discovery, microRNAs (miRNAs/miR) have been proposed as important gene regulators triggering activation or suppression pathways [15]. In this context, miRNAs can be classified as an oncogene or tumor suppressor, and their expression is not necessarily the same in different cancer types or subtypes [16]. miR-934 is an oncogene encoded as a single miRNA in the X chromosome overexpressed in BC. It is a PTEN regulator. Further, investigations suggest that miR-934 promotes cell metastasis by regulating PTEN and impacting the EMT process [17]. We identified a miRNA cluster involved in the EMT phenotype and investigated the role of miR-934 in the EMT process in different TNBC cellular lines.

## 2. Materials and Methods

### 2.1. Expression Profiles of miRNAs and mRNAs

First, from GSEA, we downloaded signatures of the mRNAs repressed during epithelial–mesenchymal transition in MCF10A cells grown at low confluency (mesenchymal phenotype) compared to those grown at high confluency (epithelial, basal-like phenotype) [18] and mRNAs downregulated during epithelial–mesenchymal transition (EMT) induced by TGFB1 [19]. Expression data of 248 differentially expressed miRNAs in ER-negative breast cancer compared to normal breast tissue were extracted from the dbDEMC database (https://www.biosino.org/dbDEMC (accessed on 29 July 2022)) with experiment ID number EXP00725 (Appendix A).

Next, we used the miRNET database to determine whether EMT-related mRNAs were the targets of miRNAs upregulated in the triple-negative breast subtype. First, we used the 2iRbase IDs of 248 overexpressed miRNAs as the input for miRNET. Next, we downloaded the experimentally validated targets for each miRNA and used the Grep function in R studio to filter the miRNAs related to the mRNAs repressed during EMT, obtaining 25 associated miRNAs.

TCGA-BRCA data from patients with triple-negative breast cancer were used to determine the expression of each reported mRNA related to EMT and miR-934. (Appendix A) Then, to analyze miR-934 expression in the nodal status and cell lines, we used cell line expression data from Ref. [20] and breast cancer samples from Ref. [20] (obtained from the UCSC Xena Browser database https://xenabrowser.net/ (accessed on 29 July 2022). Finally, to analyze the expression of miR-934 in different types of cancer, we used the miRTV database (https://mirtv.ibms.sinica.edu.tw/index.php (accessed on 29 July 2022).

### 2.2. Enrichment Analysis of Pathways and Biological Processes

We performed enrichment analysis using the bioinformatics tool enricher (https://maayanlab.cloud/Enrichr/ (accessed on 29 July 2022) and miRNA target analysis using the miRNET tool (https://www.mirnet.ca/ (accessed on 29 July 2022)). Gene set enrichment analysis (GSEA) was applied. For GSEA, we used the fgsea R package with the molecular signature database (MSigDB v.6.2) of the following gene sets: hallmark epithelial mesenchymal transition. Gene sets were considered to be significantly enriched on the basis of the normalized enrichment score (NES) >1.0.

### 2.3. Maintenance of the Breast Cancer Cell Line and Transfection of the miR-934 Inhibitor

The triple-negative breast cancer cell lines Hs-578t and MDA-MB-231 were obtained from the American Type Culture Collection and maintained in Dulbecco’s modified Eagle’s minimal medium (DMEM) supplemented with 10% bovine fetal serum and penicillin–streptomycin (50 units/mL; Invitrogen, Carlsbad, CA, USA).

For the miR-934 inhibitor transfection, 5 × 10^5^ cells were used in a 6-well plate. The following day, the cells were transfected with the miR-934 inhibitor (mirVana^®^ miRNA inhibitor, Assay ID AM12529,) at 30 nM using Lipofectamine 2000 transfection reagent (Invitrogen) in Opti-MEM Reduced Serum Medium (Life Technologies, Thermo Fisher Scientific, Grand Island, Nueva York, USA) for 48 h. Finally, we extracted the RNA to conduct RT-PCR assays and validate the knockdown.

### 2.4. RNA Isolation and qRT-PCR Analysis of miR-934 Expression

We isolated total RNA from MDA-MB-232 and HS578T cell lines using the TRIzol reagent (Invitrogen) and confirmed RNA integrity using an agarose gel. To measure miR-934 expression, we used RT-PCR microRNA assays (Applied Biosystems, Foster City, CA, USA). We used 100 ng of total RNA for reverse transcription, together with 0.15 µL of dNTPs (100 mM), 1.0 µL of reverse transcriptase Mul-tiScribe TM (50 U/µL), 1.5 µL of 10× buffer, 0.19 µL of RNase inhibitor (20 U/µL), and 4.16 µL of RNase-free water. Later, the PCR was performed with 10 µL of TaqMan master mix (Universal PCR Master mix, No AmpErase^®^ UNG, 2×), 7.67 µL of RNase-free water, and 1 µL of taqman probe PCR. The following PCR reaction was performed on an Applied Biosystems GeneAmp System 9700, Foster City, CA, USA: 95 °C for 10 min, followed by 40 cycles of 15 s at 95 °C and 40 cycles of 1 min at 60 °C. The expression of miR-934 was assessed using the Ct (2Ct) method, with U6 as a normalizer.

### 2.5. Cell Migration Assays

To assess the impact of miR-934 inhibition on MDA-MB-231 and Hs578t cell migration, we carried out a scratch/wound-healing test. Briefly, cells were cultivated to 80% confluence in a 6-well plate after being transfected with the scramble sequence as a control and the miR-934 inhibitor (30 nM). After 48 h, a vertical line was drawn using a sterile pipette tip and DMEM medium was added without supplementation. We took images at 0 h, 24 h, and 48 h and used CellProfiler 4.0, Cambridge, MA, USA, to measure the coverage area of wound healing.

### 2.6. Statistical Analysis

Each experiment was performed at least three times, and the results were presented as the mean ± SD. We used one-way analysis of variance (ANOVA) followed by Tukey’s test to compare the differences between means. A value of *p* < 0.05 was considered statistically significant.

## 3. Results

### 3.1. Identification of Differentially Expressed miRNAs in Triple-Negative Breast Cancer versus Normal Tissues Related to Epithelial–Mesenchymal Transition mRNAs

First, from the Molecular Signatures Database (MSigDB), we downloaded a list of signatures of genes downregulated during the epithelial–mesenchymal transition process. Next, we verified the expression of each gene in triple-negative breast cancer TCGA samples and found the expression to be downregulated (Figure 1A). Gene Set Enrichment Analysis (GSEA) indicates a negatively enriched set of genes in the epithelial–mesenchymal transition hallmark (Figure 1B). Next, from the dbDEMC database, we obtained the expression data of 248 miRNAs overexpressed in triple-negative breast cancer versus normal tissue. Each miRNA was examined through its mRNA targets via databases that provide experimental validation, and we obtained a total of 25 miRNAs capable of regulating the mRNAs involved in the EMT genetic signature (Figure 1C).

It is interesting to note that a significant number of mRNAs from the signatures of genes suppressed during EMT, including FOXA1, DSC3, RARRES1, PTEN, and RARRES1, have characteristics of tumor suppressors, including inhibiting hypoxia, tumor cell proliferation, and tumor cell invasion. In addition, a number of articles describe how their artificial overexpression in cell models can suppress or repress the EMT process. [21,22,23,24,25,26].

Furthermore, we hypothesize that the overexpression of mRNA signatures related to EMT may be necessary for maintaining epithelial characteristics in normal tissues, and their downregulation in breast cancer may be necessary for activating mesenchymal characteristics. Therefore, knowledge of miRNAs that regulate the genes involved in EMT and biological processes might be useful in the future for therapeutics based on EMT markers and targets.

### 3.2. Biological Processes Regulating miRNAs Related to Epithelial–Mesenchymal Transition

To determine the biological functions, we used all mRNA targets of miRNAs and EMT-related targets to carry out a functional enrichment analysis. As per our results, the identified target mRNAs are involved in many signaling pathways, such as VEGF, PI3K-AKT-mTOR, and EGR/EGFR, as well as epithelial–mesenchymal transition, apoptosis, and non-homologous end-binding mechanisms (Figure 2A). In addition, pathways related to EMT genes are involved in signaling pathways such as FoxO, TGF- β, p53, and apoptosis (Figure 2B). Interestingly, gene ontology (GO) analysis demonstrates that the process characteristics of EMT, such as cell adhesion, loss of apical–basal polarity, and the degradation of the basal extracellular, are involved in biological processes, cellular components, and molecular functions (Figure 2C). Previous studies have also reported that these pathways could play a relevant role in the EMT process in cancer. For example, in hepatocellular carcinoma (HCC), FoxO 1 overexpression suppresses HCC invasion and migration by reversing TGF-β-induced EMT [27].

To carry out a more complete analysis, we determined the relationship between the pathways and the mRNAs (Figure 3). As per the results, several pathways related to EMT signaling pathways that inhibit EMT by activating Hippo–YAP signaling [28] are associated with a set of genes. Examples are SGK1, PTEN, STAT3, FBXO31, and p53 pathways that prevent EMT by repressing ZEB1 and ZEB2 expression [29]. Apoptosis genes, such as BIRC3, promote EMT in liver cancer [30]. Human T-cell leukemia virus 1 infection is related to the PTEN that suppresses epithelial–mesenchymal transition and cancer stem cells [24]. In conclusion, functional enrichment analysis of miRNAs and their target mRNAs provided us with insights into their functional roles in EMT.

### 3.3. miRNA–mRNA Interaction Networks Show miR-934 as a Possible Key Regulator of Epithelial–Mesenchymal Transition

In particular, we observed the coregulation of upregulated miRNAs and their respective downregulated mRNAs in TNBC and also observed the enrichment in the nodes of mRNAs subjected to regulation by several miRNAs, such as PTEN, KLF6, GAS1, HMGA2, and BTG1. Several miRNAs have multiple mRNAs; for example, miR-1277-5p has BCL6, PTEN, and TP53INP1. While miR-454-3p targets STAT3, MAFB, and BCL6, miR-142-3p can bind to SGK1 and MYH9, among others, as shown in Figure 4. In a coregulatory network, a single mRNA can interact with more than one miRNA, and one miRNA is capable of interacting with several mRNAs. Interestingly, miR-934 is capable of regulating multiple targets, including PTEN, TP53INP1, HMGA2, EGR2, EIF4B, and INSIG1. Previous research has shown that miR-934 is an important regulator in EMT, and our results are consistent with this evidence, showing that miR-934 has binding sites for different ETM-related mRNAs, such as PTEN, EGR2, FOSL2, and MGA2. The interaction of miR-934 and its target mRNAs is highlighted in red in Figure 5.

### 3.4. A Landscape of miR-934 Expression in Different Types of Cancers

Given the possible importance of miR-934 in cancer, we decided to carry out an expression analysis of this microRNA in different types of cancers, finding that it is overexpressed in relation to normal tissue in thyroid cancer (THCA), breast cancer (BRCA), uterine corpus endometrial carcinoma (UCEC), lung adenocarcinoma (LUAD), and bladder urothelial carcinoma (BLCA). However, we found that in kidney cancers, such as kidney renal clear cell carcinoma (KIRC) and kidney chromophobe (KICH), miR-934 is overexpressed in normal tissues. (Figure 4)

### 3.5. The Upregulation of miR-934 Is Specific to the Molecular Subtype of Triple-Negative Breast Cancer

The invasion of cancer cells into lymph nodes has been recognized as the primary early metastatic pathway. Currently, the EMT mechanism can be crucial in the spread of cancer, involving several factors related to lymph node metastasis. Interestingly, we identified that miR-934 overexpression is higher in triple-negative breast cancer patients with lymph nodes + compared to luminal-type breast cancer patients with lymph nodes + (Figure 6A). This points to the possible function of miR-934 in the EMT process and its affinity for the triple-negative molecular subtype. Furthermore, among the many triple-negative breast cancer cell lines, HS578t, MDA-MB-231, and BT549 cells show a greater expression of miR-934 and luminal-type breast cancer cells (Figure 6B).

Subsequently, we used TCGA BRCA data to analyze the expression levels of miR-934 in patients with triple-negative breast cancer and its associated target mRNAs (PTEN, EGR2, ANK3, ZBTB16, TCF3, CCNG1, FOSL2, BTG1, HMGA2, TP53AIP1, and INSIG1) and found that the overexpression of miR-934 significantly discriminates triple-negative molecular subtype from normal breast tissue (Figure 6C). Additionally, an inverse effect is observed on the expression of its different target mRNAs, the effect being clearer for PTEN and EGR2, which are downregulated in triple-negative breast cancer samples (Figure 6C). Expression analysis using PAM50 criteria shows that miR-934 is overexpressed only in triple-negative breast cancer, compared to luminal A, luminal B, and HER2 subtypes (Figure 6D). These findings lead us to speculate about the specificity of miR-934 in triple-negative subtypes. Interestingly, PTEN and EGR2 expression (Figure 6E,F) is downregulated in the triple-negative subtype compared to the other subtypes, indicating a possible mechanism of negative gene expression regulation by miR-934.

### 3.6. miR-934 Inhibition Partly Suppresses the Migration Capacity in Triple-Negative Breast Cancer Cells

To corroborate the miR-934 functions in cancer hallmarks, we inhibited its expression in TNBC cells such as HS-578T and MDA-MB-231 through a miRNA inhibitor. The miR-934 inhibition was demonstrated by an RT-PCR assay in both cell lines (Figure 7E). Interestingly, HS578T overexpresses miR-934 when compared to MDA-MB-231. To determine whether these expression differences might affect the migration capacity after miR-934 inhibition, we performed a wound-healing assay to study cell migration and found that the wound coverage in HS578T and MDA-MB-231 is greater in control and scramble cells compared to cells treated with the miR-934 inhibitor, where it is observed that the wound coverage is similar at 24 and 48 h (Figure 7A–C). Our results suggest that the inhibition of miR-934 in TNBC cell lines is capable of partly inhibiting the migration capacity.

## 4. Discussion

TNBC is the most challenging subtype of breast cancer and does not benefit from the existing targeted therapies. In the present study, we used bioinformatics approaches to assess the miRNAs involved in the EMT pathways and the GO process, which may explain the ability of miRNAs to regulate EMT and the invasive features of TNBC. For example, miR-1297 inhibition led to the upregulation of PTEN and phenotypically suppresses invasion and migration, in addition to suppressing the AKT/ERK pathway [32].

Our results also demonstrated that the identified target mRNAs are involved in many signaling pathways, such as VEGF, PI3K-AKT-mTOR, and EGR/EGFR, as well as epithelial–mesenchymal transition, apoptosis, and non-homologous end-binding mechanisms. Furthermore, the Hippo signaling pathway and NF-κβ have been described to be involved in EMT, invasion, and metastasis [33,34]. These functional annotation results could help us take advantage of the potential role of miRNAs in EMT by regulating their targets in breast cancer.

Interestingly, we found that miR-943 regulates multiple-target mRNAs of a genetic signature that are repressed during the EMT process, which could partly explain the low expression of these mRNAs in the triple-negative subtype (Figure 5). Interestingly, miR-934 is overexpressed only in triple-negative subtypes (Figure 6D). We speculate that miR-934 overexpression is TNBC-specific and could function as a possible biomarker.

A possible explanation is an inverse correlation between miR-934 expression and hormonal receptors and HER2/neu status in breast cell lines and tumors (Figure 6B). These data agree with a previous study that reports that ER-α is a target of miR-934, suggesting the inhibition of this receptor in triple-negative breast cancer [35,36].

Additionally, mRNA targets of miR-934 show an inverse correlation and a lower expression in triple-negative breast cancer. The interaction network suggests that miR-934 plays an important role in the development of EMT in breast cancer and has binding sites for multiple mRNAs. In particular, PTEN and EGR2 play important roles in EMT inhibition, and interestingly our analysis shows that PTEN and EGR2 have the lowest expression in the triple-negative subtype compared to luminal and HER2+ (Figure 6C,D). With these findings, we suggest miR-934–PTEN–EGR2 axis regulation for inducing EMT in highly invasive cells, indicating possible future therapeutic targets. However, more studies and different models are needed.

## 5. Conclusions

Analysis of sets of miRNAs helps determine potential individual therapeutic targets in exacerbated cancer hallmarks. We found that 25 miRNAs that regulate mesenchymal genes are overexpressed in patients with TNBC. The expression level of miR-934 is specific to the molecular subtype of triple-negative breast cancer and modulates a set of related epithelial–mesenchymal genes. In addition, miR-934 inhibition in TNBC cell lines inhibits the migratory abilities of tumor cells, which makes this miRNA an excellent candidate for further study.

## Figures and Tables

**Figure 1 cells-12-00834-f001:**
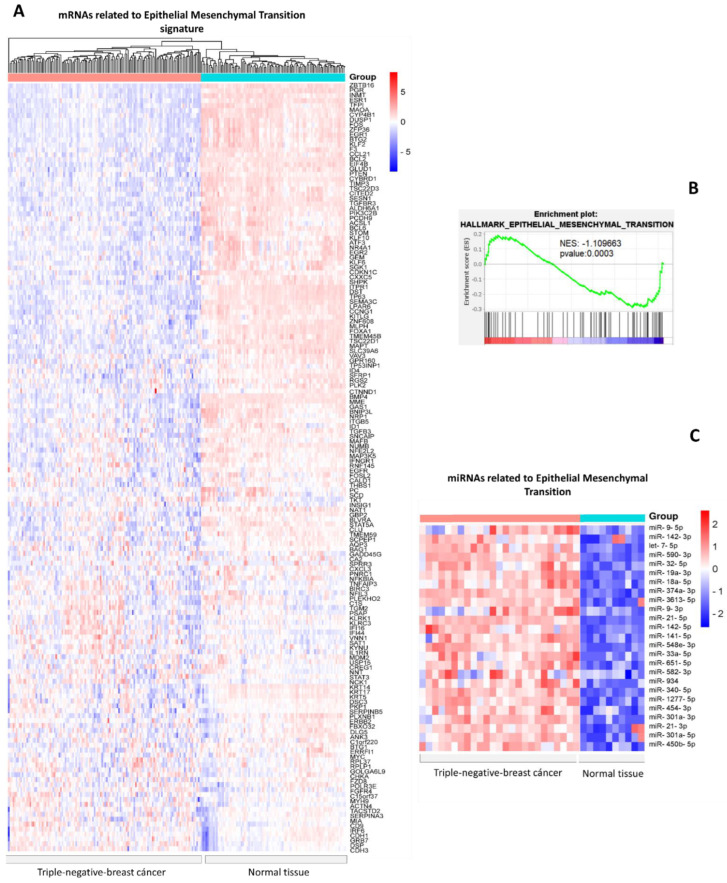
Expression profile of miRNAs-mRNAs related to epithelial mesenchymal transition in triple-negative breast cancer vs. normal breast tissue. (**A**) Heat map with hierarchical clustering of mRNAs from TCGA (blue: Normal tissues, red; breast cancer triple-negative). (**B**) GSEA analysis of gene set of epithelial–mesenchymal genes. (**C**) Heat map of miRNAs expression in breast cancer vs. normal tissues. Red shows the miRNAs in normal tissue and blue shows triple-negative breast cancer samples.

**Figure 2 cells-12-00834-f002:**
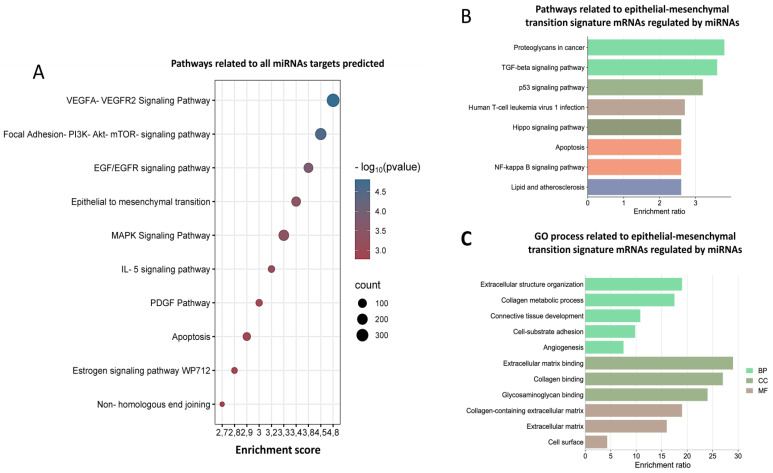
Functional enrichment analysis of mRNAs regulated by selected 25 miRNAs. (**A**) Pathways related to all miRNAs target’s prediction; the dots represent the p-value of pathways. (**B**,**C**) Pathways and GO process related to mRNAs of MTE signature regulated by 25 miRNAs, the bar graphs represent the enrichment score of the genes involved in each biological process in pathways and gene ontology (GO).

**Figure 3 cells-12-00834-f003:**
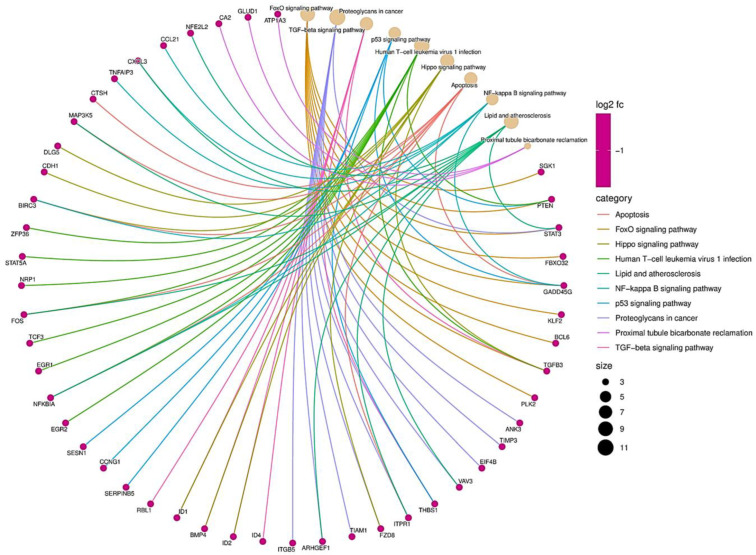
Biological function of miRNAs related to epithelial–mesenchymal transition genes. The size of the dots is proportional to the number of genes related to the GO term. The pink dots represent genes, while the brown dots represent biological processes. The different colored strings represent the relationship between the biological function and the gene.

**Figure 4 cells-12-00834-f004:**
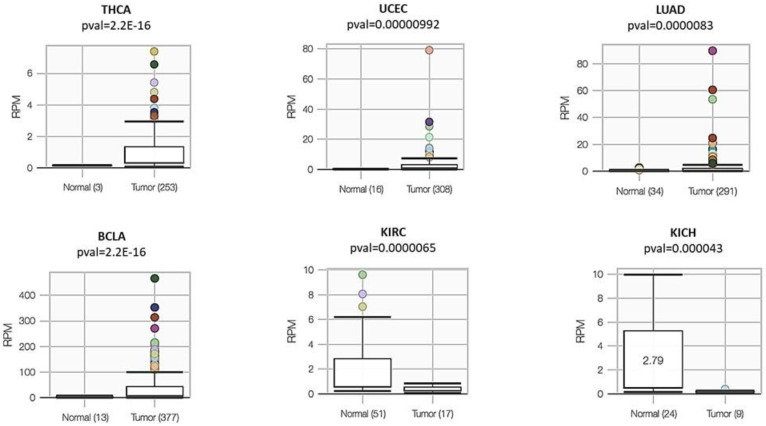
TCGA data showing expression of miR-934 in various cancer types vs. normal tissue from TCGA version 18.0 via miRTV database. The color dots represent outlier values in expression data of each type of cancer.

**Figure 5 cells-12-00834-f005:**
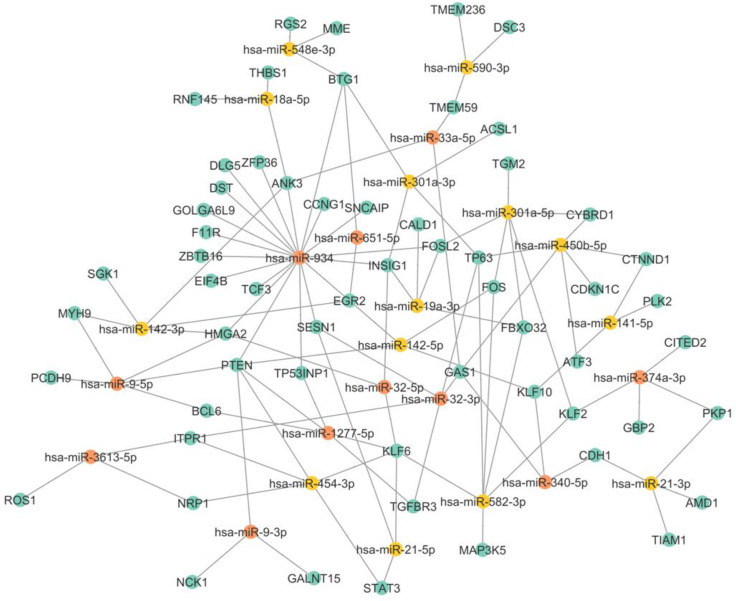
miRNA-mRNA interaction network. Yellow dots represent each of the miRNAs, while blue dots represent white mRNAs. The dotted lines represent the interaction between both molecules. The red dotted lines highlight the interactions between the miR-934 and its targets.

**Figure 6 cells-12-00834-f006:**
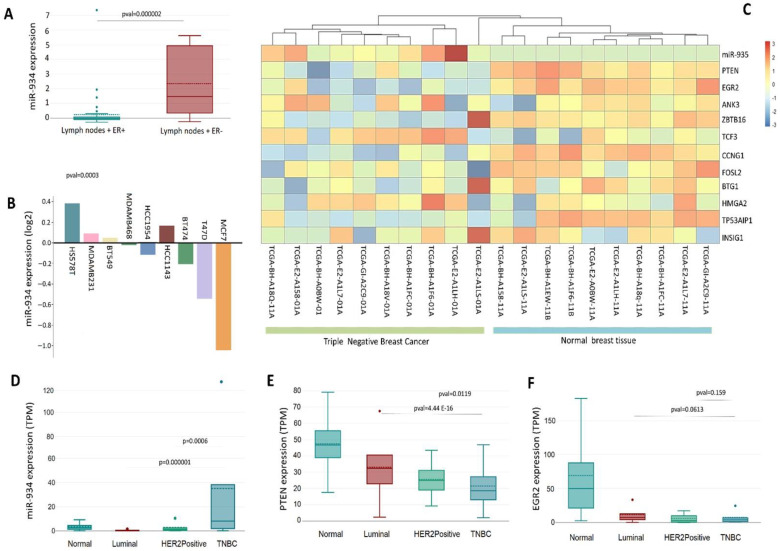
Differential expression of miR-934, EGR2, and PTEN in breast cancer tissue samples compared to normal tissue. (**A**) Expression of miR-934 related to nodal status + in ER+ and ER−, using Chin et.al dataset [20]. (**B**) Expression of miR-924 in breast cancer cell line using Neve et.al dataset [31]. (**C**) Profile expression of miR-934 and target genes in triple-negative breast cancer and normal tissue using TCGA-BRCA dataset. (**D**–**F**) Expression profile of miR-934, PTEN, and EGR2 by molecular subtype with TCGA data set. Dots are outliers in the boxplox data from TCGA.

**Figure 7 cells-12-00834-f007:**
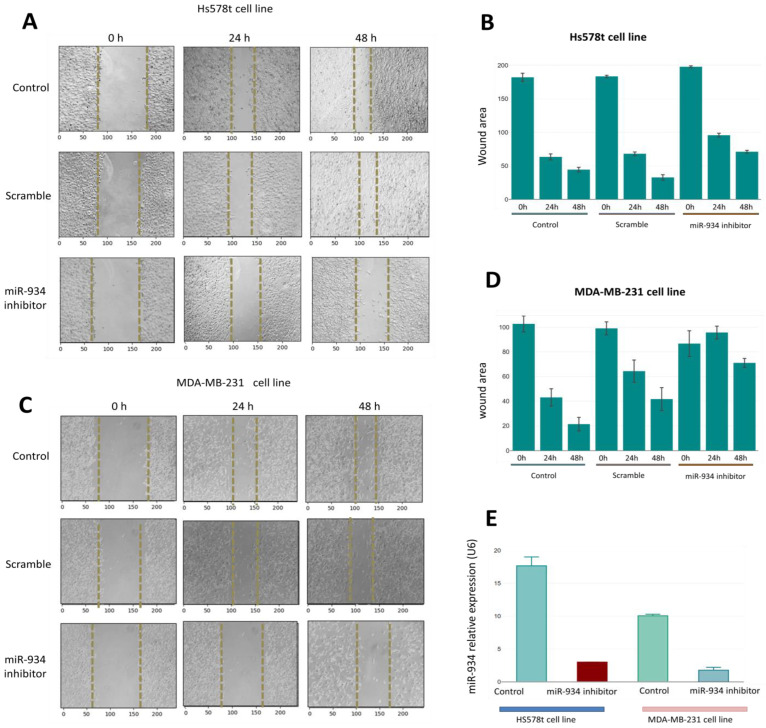
Knockdown of miR-934 suppresses cell migration in TNBC cell lines. (**A**–**C**) Invasion and migration were evaluated in Hs-578t and MDA-MB-231 cells treated with the miR-934 inhibitor, scrambled and untransfected over the course of 0 h, 24 h, and 48 h. In (**B**–**D**), the wound coverage in the cells in the different conditions after time is shown. (**E**) The expression level of miR-934 in cell line Hs-578t andMDA-MB-231 with and without miR-934 inhibitor transfection.

## Data Availability

Not applicable.

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
