# Peer review of "Bioinformatic miRNA-mRNAs Analysis Revels to miR-934 as a Potential Regulator of the Epithelial–Mesenchymal Transition in Triple-Negative Breast Cancer"

_cells, 2023, doi:10.3390/cells12060834_

Round 1

Reviewer 1 Report

The authors present a dataset driven study to identify novel miRNAs involved in breast cancer progression and metastasis. In order to do to so they focus on miRNAs particularly involved in the EMT process in TNBC and detect miRNA-934 as a specific miRNA for the TNBC molecular subtype. In a short functional segment the effect of an undefined miRNA-934 commercial inhibitor is tested on two TNBC cell lines and their migratory ability under treatment is investigated.

The study as it is presented is scientifically sound but lacks novelty. Lu et al. (2021) described the effect of miR-934 on BC metastasis in some detail and showed already PTEN being affected as well as the impact on EMT. Therefore it is difficult to understand the reasoning of the authors in just providing a small functional part that seems very repetitive of that work.

The use of miR-934 as a potential TNBC marker is however somewhat interesting but should have been explored in more detail. Here the authors just skim the surface of a possible study. The expression data should be verified using different methods and also compared to other types of cancer as well as normal tissues not just from the breast. There is an interesting opportunity here to focus on miR-934 as a marker in this reviewers opinion.

The functional data fall somewhat short of definitive proof. The undefined “inhibitor” needs clarification. An alternative approach with antagomirs should be added to define migratory effects, that seem variable between the two cell lines.

All in all there are interesting points to this study but they should have been developed further to merit publication in Cells.

Author Response

Dear Reviewer:

Good afternoon!
To thank you for the time you took to review our proposal. They were very helpful and improved the quality of the article.
Please find attached the response to your suggestions. 
Best regards!!

Reviewer 2 Report

 One major issue of the paper is that the Methods section is not documented in detail. For example, what are the study populations that gave rise to the data retrieved from dbDEMC and MSigDB. Since mRNA expression and miRNA expression were retrieved from different databases, it is unclear how these two data sources can be unified. Additionally, there was no description as to how the authors run miRNET and how many mRNAs and miRNAs were kept vs. removed for functional enrichment analysis? How did you determine the EMT related genes?

Is there a formal statistical test to determine the miR-934 is the most promising candidate?

In section 3.5, how do you explain the results? Why were these specific two cell lines selected in experimental validation?

In line 73, you mentioned “repressed” genes. Did you only consider repressed genes and overexpressed miRNA? Why was the opposite relationship omitted?

There are a lot of typos and missing references. For example, in lines 62-63 “Since its discovery microRNAs (miRNAs/miR) was proposed such as important gene regulators triggering the activation or suppression pathways {ref}.”.

The figures are not self-explanatory (missing legend titles etc.). For example, Figure 1A and 1C are confusing as the color was reversed to indicate cancer vs. normal tissues.

Author Response

(The authors gave the same response as above.)

Round 2

Reviewer 1 Report

The authors tried to improve the manuscript according to suggestions and to explain to this reviewer why the data presented here merit publication. I still feel, the study lacks novelty and could profit from additonal work as suggested in the first round of reviews but I will leave this final assessment to the editorial staff. 

Reviewer 2 Report

The authors have addressed all my previous comments and concerns.